# The Global Spectra-Trait Initiative: A database of paired leaf spectroscopy and functional traits associated with leaf photosynthetic capacity

Julien Lamour[1], Shawn P. Serbin[2], Alistair Rogers[3], Kelvin T. Acebron[4], Elizabeth Ainsworth[5], Loren P. Albert[6], Michael Alonzo[7], Jeremiah Anderson[2], Owen K. Atkin[8,9], Nicolas Barbier[10], Mallory L. Barnes[11], Carl J. Bernacchi[12,5], Ninon Besson[10,13], Angela C. Burnett[14], Joshua S. Caplan[15], Jérôme Chave[1], Alexander W. Cheesman[16], Ilona Clocher[10,13], Onoriode Coast[17], Sabrina Coste[13], Holly Croft[18,19], Boya Cui[20], Clément Dauvissat[10], Kenneth J. Davidson[21], Christopher Doughty[22], Kim S. Ely[3], John R. Evans[23], Jean-Baptiste Féret[24], Iolanda Filella[25], Claire Fortunel[10], Peng Fu[26], Robert T. Furbank[23], Maquelle Garcia[6], Bruno O. Gimenez[27,28], Kaiyu Guan[29,30], Zhengfei Guo[31], David Heckmann[32], Patrick Heuret[10], Marney Isaac[20], Shan Kothari[33], Etsushi Kumagai[34], Thu Ya Kyaw[22], Liangyun Liu[35], Lingli Liu[36], Shuwen Liu[37,31], Joan Llusià[25], Troy Magney[38], Isabelle Maréchaux[10], Adam R. Martin[20], Katherine Meacham-Hensold[39], Christopher M. Montes[12,5], Romà Ogaya[25], Joy Ojo[17], Regison Oliveira[28], Alain Paquette[40], Josep Peñuelas[25], Antonia Debora Placido[28], Juan M. Posada[41], Xiaojin Qian[42], Heidi J. Renninger[43], Milagros Rodriguez-Caton[44,38], Andrés Rojas-González[45], Urte Schlüter[46], Giacomo Sellan[13], Courtney M. Siegert[43], Viridiana Silva-Perez[23], Guangqin Song[31], Charles D. Southwick[6], Daisy C. Souza[28], Clément Stahl[13], Yanjun Su[36], Leeladarshini Sujeeun[20], To-Chia Ting[47], Vicente Vasquez[48], Amrutha Vijayakumar[17], Marcelo Vilas-Boas[28], Diane R. Wang[47], Sheng Wang[49,29], Han Wang[50], Jing Wang[51], Xin Wang[36], Andreas P.M. Weber[46,52], Christopher Y.S. Wong[53], Jin Wu[31,54,55], Fengqi Wu[36], Shengbiao Wu[56], Zhengbing Yan[36,57,58], Dedi Yang[59], Yingyi Zhao[31]

[1]Centre de Recherche sur la Biodiversité et l'Environnement (CRBE), Université de Toulouse, CNRS, IRD, Toulouse INP, Université Toulouse 3 – Paul Sabatier (UT3), Toulouse, France
[2]Biospheric Sciences Laboratory (BSL), Code 618 NASA Goddard Space Flight Center 8800 Greenbelt Road Greenbelt, MD 20771 USA
[3]Climate and Ecosystem Sciences Division, Lawrence Berkeley National Laboratory, Berkeley CA 94720, USA
[4]Smithsonian Environmental Research Center, Edgewater MD 21403
[5]University of Illinois Urbana, Champaign, Urbana, IL 61801
[6]Oregon State University Department of Forest Ecosystems & Society, Corvallis, OR 97333
[7]Department of Environmental Science, American University, Washington, DC 20016
[8]ARC Centre of Excellence in Plant Energy Biology, Research School of Biology, The Australian National University, Canberra, ACT 2601, Australia
[9]Division of Plant Sciences, Research School of Biology, The Australian National University, Canberra, ACT 2601, Australia
[10]AMAP, Univ Montpellier, CIRAD, CNRS, INRAE, IRD, Montpellier, France
[11]O'Neill School of Public and Environmental Affairs, Indiana University
[12]USDA-ARS Photosynthesis Research Unit, Urbana, IL, USA 61801
[13]UMR EcoFoG, AgroParisTech, Cirad, CNRS, INRAE, Université des Antilles, Université de la Guyane, Kourou, France
[14]Advanced Research + Invention Agency, London, UK
[15]Department of Architecture & Environmental Design, Temple University, Ambler PA 19010
[16]College of Science & Engineering, James Cook University, Cairns, Queensland 4878, Australia
[17]School of Environmental and Rural Science, University of New England, Armidale, NSW 2351, Australia
[18]Plants, Photosynthesis and Soil, School of Biosciences, University of Sheffield, South Yorkshire, UK
[19]School of Biosciences, Institute for Sustainable Food, University of Sheffield, South Yorkshire, UK
[20]Department of Physical and Environmental Sciences, University of Toronto Scarborough, Canada, M1C 1A4
[21]American Forests, Washington, DC 20005, USA
[22]School of Informatics, Computing, and Cyber Systems, Northern Arizona University, Flagstaff, AZ, USA
[23]ARC Centre of Excellence for Translational Photosynthesis, Research school of Biology, Australian National University, Canberra, ACT 2601, Australia
[24]TETIS, INRAE, AgroParisTech, CIRAD, CNRS, Université Montpellier, Montpellier, France

[25]Center for Ecological Research and Forestry Applications (CREAF) - National Research Council (CSIC), Edifici C, Universitat Autonoma de Barcelona, 08193 Bellaterra, Barcelona

[26]School of Plant, Environmental and Soil Sciences, Louisiana State University and Louisiana State University AgCenter, Baton Rouge, LA 70803

[27]Department of Geography, University of California - Berkeley (UCB), 507 McCone Hall #4740, Berkeley, CA 94720, USA

[28]Forest Management Laboratory, National Institute of Amazonian Research (INPA), Av. Andre Araújo, 69060-082, Manaus-AM, Brazil

[29]Agroecosystem Sustainability Center, Institute for Sustainability, Energy, and Environment, University of Illinois Urbana-Champaign, Urbana, IL 61801, USA

[30]Department of Nature Resources and Environmental Sciences, College of Agricultural, Consumer and Environmental Sciences, University of Illinois Urbana-Champaign, Urbana, IL 61801, USA

[31]School of Biological Sciences, University of Hong Kong, Pokfulam Road, Hong Kong, China

[32]Bayer Crop Science, 40789 Monheim am Rhein, Germany

[33]Department of Renewable Resources, University of Alberta, Edmonton, AB T6G 2E3, Canada

[34]Institute for Agro-Environmental Sciences, NARO, Tsukuba, Ibaraki 305-8604, Japan

[35]Aerospace Information Research Institute, Chinese Academy of Sciences, Beijing 100094, China

[36]State Key Laboratory of Vegetation and Environmental Change, Institute of Botany, Chinese Academy of Sciences, Xiangshan, Beijing 100093, China

[37]Department of Forest and Wildlife Ecology, University of Wisconsin-Madison, 1630 Linden Drive, Madison, WI 53706, USA

[38]Department of Plant Sciences, University of California Davis, USA

[39]Carl R Woese Institute for Genomic Biology, University of Illinois Urbana Champaign, Urbana, Illinois, USA

[40]Centre for Forest Research, Université du Québec à Montréal, Montréal, Canada

[41]Biology Department, Carrera 24 # 63C - 69, Universidad del Rosario, Bogotá, Colombia

[42]School of Internet of Things, Nanjing University of Posts and Telecommunications, Nanjing 210003, China

[43]Department of Forestry, Mississippi State University, Mississippi State, MS 39762

[44]Institute for Snow, Glaciers and Environmental Research, IANIGLA-CONICET, Argentina

[45]Laboratorio de Ecología Funcional y Ecosistemas Tropicales (LEFET), Escuela de Ciencias Biológicas, Facultad de Ciencias Exactas y Naturales, Universidad Nacional, Heredia, Costa Rica

[46]Institute for Plant Biochemistry, Heinrich Heine University Düsseldorf, Germany

[47]Agronomy Department, Purdue University, West Lafayette, IN, United States

[48]University of Florida. School of Forestry, Fisheries and Geomatics

[49]Department of Agroecology, Aarhus University, Aarhus, Denmark, 8000

[50]Ministry of Education Key Laboratory for Earth System Modelling, Department of Earth System Science, Tsinghua University, Beijing 100084, China

[51]School of Ecology, Shenzhen Campus of Sun Yat-sen University, Shenzhen, Guangdong, 518107, China

[52]Cluster of Excellence on Plant Sciences (CEPLAS), Düsseldorf, Germany

[53]Faculty of Forestry and Environmental Management, University of New Brunswick, Fredericton NB E3B 5A3, Canada

[54]Institute for Climate and Carbon Neutrality, University of Hong Kong Kong, Pofulam Road, Hong Kong, China

[55]State Key Laboratory of Agrobiotechnology, Chinese University of Hong Kong, Sha Tin, Hong Kong, China

[56]Future Urbanity and Sustainable Environment (FUSE) Lab, Division of Landscape Architecture, Department of Architecture, Faculty of Architecture, The University of Hong Kong, Hong Kong, China

[57]China National Botanical Garden, Beijing 100093, China

[58]University of Chinese Academy of Sciences, Yuquanlu, Beijing 100049, China

[59]Environmental Sciences Division and Climate Change Science Institute, Oak Ridge National Laboratory, Oak Ridge, TN 37830, USA

*Correspondence to*: Julien Lamour (jlamour.sci@gmail.com)

**Abstract.** Accurate assessment of leaf functional traits is crucial for a diverse range of applications from crop phenotyping to parameterizing global climate models. Leaf reflectance spectroscopy offers a promising avenue to advance ecological and agricultural research by complementing traditional, time-consuming gas exchange measurements. However, the development of robust hyperspectral models for predicting leaf photosynthetic capacity and associated traits from reflectance data has been hindered by limited data availability across species and environments. Here we introduce the Global Spectra-Trait Initiative (GSTI), a collaborative repository of paired leaf hyperspectral and gas exchange measurements from diverse ecosystems. The GSTI repository currently encompasses over 7500 observations from 397 species and 41 sites gathered from 36 published and unpublished studies, thereby offering a key resource for developing and validating hyperspectral models of leaf photosynthetic capacity. The GSTI database is developed on GitHub (https://github.com/plantphys/gsti) and published to ESS-DIVE https://data.ess-dive.lbl.gov/datasets/doi:10.15485/2530733, Lamour et al., 2025). It includes gas exchange data, derived photosynthetic parameters, and key leaf traits often associated with traditional gas exchange measurements such as leaf mass per area and leaf elemental composition. By providing a standardized repository for data sharing and analysis, we present a critical step towards creating hyperspectral models for predicting photosynthetic traits and associated leaf traits for terrestrial plants.

## 1 Introduction

The structural, chemical, and physiological properties of plants, commonly known as plant traits or plant functional traits (Violle et al., 2007), directly impact leaf, root, whole-plant, and ecosystem functioning as well as their responses to global and environmental change (Kattge et al., 2020; Reich et al., 1997). Leaf traits have become increasingly important for crop phenotyping, selection for breeding, precision agriculture, biodiversity conservation and for modeling plant and ecosystem processes using land surface models (Bjorkman et al., 2018; Fu et al., 2022; Meacham-Hensold et al., 2020; Xiong and Flexas, 2018). Of note, photosynthetic traits and associated traits such as leaf nitrogen content are of particular value to these efforts (Walker et al., 2014). This is in part due to their important role in determining leaf-to-global scale fluxes of carbon, water, and energy, as well as the outsized impact photosynthetic traits have on key model outputs (Bonan et al., 2011; Ricciuto et al., 2018; Rogers, 2014; Rogers et al., 2017a). Indeed, uncertainty in photosynthetic traits has been shown to have a greater impact on future climate projections by land surface models than the uncertainty associated with global change (Liu et al., 2024; Stinziano et al., 2018). Unfortunately, the global coverage of functional traits, especially physiological traits associated with photosynthesis, is hampered by logistical constraints that limit their spatial, temporal, and environmental coverage, as well as the diversity of species covered (Feng and Dietze, 2013; Keenan and Niinemets, 2016; Schimel et al., 2015).

Photosynthetic traits are traditionally inferred from a biochemical model of photosynthesis (Farquhar et al., 1980; Yin et al., 2021) calibrated using leaf-level gas exchange techniques (Busch et al., 2024; Long and Bernacchi, 2003). The key traits governing photosynthesis are the maximum carboxylation capacity of the enzyme rubisco ($V_{cmax}$), the maximum potential rate of electron transport ($J_{max}$), the maximum capacity for triose phosphate utilization ($TPU$), and the rate of mitochondrial

respiration in the dark ($R_{dark}$), all expressed at a reference temperature, typically 25°C. Estimation of $V_{cmax}$, $J_{max}$, and $TPU$ is achieved by measuring the response of net $CO_2$ assimilation ($A$) to changes in the intercellular $CO_2$ concentration ($C_i$), commonly known as an A-Ci curve. $R_{dark}$ is estimated from the rate of $CO_2$ released from dark-acclimated leaves. The A-Ci

and $R_{dark}$ protocols typically take ~45 minutes each to complete (Busch et al., 2024; Rogers et al., 2017b). Although alternative gas exchange approaches are faster (De Kauwe et al., 2016; Saathoff and Welles, 2021; Stinziano et al., 2017; Tejera-Nieves et al., 2024), these approaches may also require a significant stabilization period prior to measurement (Burnett et al., 2019). Photosynthetic traits are also inferred from their empirical relationships with other leaf traits that are easier to measure, such as leaf mass per area (LMA) or leaf nitrogen (N), phosphorous (P), or chlorophyll content (Croft et al., 2017; Domingues et

al., 2010; Walker et al., 2014). Although these relationships are useful for scaling photosynthetic traits across broader scales (Rogers et al., 2017a), they are often site- and species-specific (Feng and Dietze, 2013; Keenan and Niinemets, 2016; Yan et al., 2021).

The structural, chemical, and physiological properties of a leaf impact its optical properties (Serbin and Townsend, 2020), 2020). Measuring the optical properties of a leaf, i.e., its reflectance across multiple wavelengths using spectroscopy, has

emerged as a tool for retrieving leaf traits, including photosynthetic and many other traits, in a non-destructive and high-throughput manner (Coast et al., 2019; Lamour et al., 2021; Serbin et al., 2012; Wu et al., 2025, 2019; Yan et al., 2021). A variety of statistical approaches can be used to retrieve leaf trait values from leaf reflectance measurements. These include simple indices (Gitelson et al., 2003; Rouse et al., 1974), which relate specific wavelengths to traits of interest, to more sophisticated multivariate techniques like partial least square regression (PLSR) and deep learning (Burnett et al., 2021;

Furbank et al., 2021; Wold et al., 2001) which are capable of using information from numerous wavelengths to predict traits. Such approaches are effective at predicting chemical and structural leaf traits (Kothari et al., 2023; Serbin et al., 2019), as well as photosynthetic traits (Heckmann et al., 2017; Montes et al., 2022; Wu et al., 2025, 2019). However, a significant bottleneck for the estimation of traits from spectra is the development of models that can be used broadly across environments and species where ideally, the broadest possible combination of structural, chemical, physiological and optical traits can be represented by

a single model. Often, models developed for a given biome, species, or growth environment perform poorly when used with new species or in a different environment (Burnett et al., 2021; Coast et al., 2019; Lamour et al., 2021; Meacham-Hensold et al., 2019). It is therefore increasingly clear that robust models require training data that covers the widest possible range of values for a given trait, and the widest possible diversity in leaf optical properties that are correlated with that trait.

Here we introduce the Global Spectra-Trait Initiative (GSTI), a collaborative repository of paired leaf reflectance and gas

exchange measurements from diverse ecosystems. Unlike existing databases for leaf traits (EcoSIS - Spectral Library, 2025; Kattge et al., 2020), the GSTI is specifically focused on paired datasets of key leaf traits linked to leaf spectroscopy taken on the same sample and where that link is preserved with unique identifiers. Furthermore, the GSTI includes the raw data that underlies trait estimation, in this case, leaf-level gas exchange, leaf structural and compositional trait data, and leaf reflectance spectra data. Furthermore, the GSTI provides an open workflow to process raw reflectance and gas exchange data ensuring

uniform and reproducible data processing and interpretation. The GSTI therefore enables and maximizes further reuse of the

data, facilitates the ongoing refinement of spectra-trait models as new datasets are incorporated, supports comparisons of diverse statistical approaches, and permits the reinterpretation of stored gas exchange data using alternative or new photosynthetic models (Busch et al., 2018; Johnson and Berry, 2021; Márquez et al., 2021).

## 2 Methods

### 2.1 Data sources

The primary aim of the GSTI is to collate paired leaf reflectance and gas exchange data that can be used to estimate the key photosynthetic traits, namely $V_{cmax}$, $J_{max}$, $TPU$ and $R_{dark}$, and associated functional traits. Data from multiple biomes, species, growth conditions, and plant types, including crops and wild species, have been collated into a single, open database with the aim to capture a wide variety of leaf traits and their associated optical properties.

The GSTI database primarily focuses on gas exchange data measured with the conventional steady state A-Ci curve protocol (Busch et al., 2024), where photosynthesis is modulated by changing the $CO_2$ concentration at the leaf surface using pre-defined increments. However, the database also includes data measured with the simplified "one point" method (Burnett et al., 2019; De Kauwe et al., 2016), where the photosynthesis rate is measured at saturating irradiance and ambient $CO_2$, as well as A-Ci curves measured with the non-steady state dynamic assimilation technique (Saathoff and Welles, 2021; Tejera-Nieves et al., 2024).

Estimation of photosynthetic parameters from A-Ci curves depends on the choice of the biochemical model of photosynthesis, its parametrization, and the statistical procedure of model fitting, all of which are likely to vary between studies (Rogers et al., 2017a). Therefore, a requirement of data contributors was to provide the gas exchange measurements so that data could be used to estimate photosynthetic traits using the same biochemical model, with the same assumptions and parameters. Each dataset was curated to ensure that the data were organized, standardized and free of errors before inclusion in GSTI. Restricting gas exchange data contributions to the raw gas exchange measurements vs. simply the fitted parameters (e.g., $V_{cmax}$, $J_{max}$, $TPU$) avoided issues of mixed fitting approaches and assumptions, which would have increased uncertainties in the final spectral models.

The other primary data contributions to the GSTI database were measurements of leaf reflectance. The reflectance data were either collected with a leaf clip (i.e., a contact probe with an articulating backplate) or with an integrating sphere. Reflectance data spanning the range from 400 to 2500 nm were preferred but data from reduced wavelength ranges (e.g., only the visible spectrum through near-infrared) were also included.

In general, paired measurements of gas exchange and reflectance on the same leaves were preferred, to avoid leaf-to-leaf variation in leaf traits. However, measurements on similar or "analog" leaves were also accepted in the GSTI database, provided gas exchange and reflectance were taken on a similar leaf of identical age and appearance.

Where available, other leaf traits complementary to gas exchange data were also added to GSTI. These included most of the traits correlated with photosynthetic capacity (Domingues et al., 2010; Walker et al., 2014; Wang et al., 2022), such as LMA, and leaf N and P content, as well as leaf water content (LWC).

For all contributed datasets, the materials and methods used for the study were described following standard scientific requirements and are captured in the GSTI database metadata. This includes descriptions of the site, plant, protocols, and equipment. Only Open Data (CC BY 4.0, https://creativecommons.org/licenses/by/4.0/deed.en) from published and unpublished studies were included in the GSTI database. These are free to use, reuse, share and adapt without restriction. By limiting the GSTI database to Open Data only, it is our philosophy that the GSTI database will also facilitate scientific collaboration, transparency, and reproducibility, as well as accelerate discovery and understanding in the areas of plant science.

## 2.2 Repository organisation

### 2.2.1 Overall organisation

The overall design philosophy of the GSTI database is to provide an easy, accessible, and interpretable repository of paired spectroscopy and leaf functional traits. The GSTI repository and associated database are available on GitHub (https://github.com/plantphys/gsti). The repository was designed to be flexible enough to accept data from many studies and support raw data from a range of instruments in a free format. However, because the goal of GSTI is to provide a means of synthesizing and standardizing data into common formats, units, and metadata, we provide a small but strict set of requirements for data contributors to make it possible to process and then curate the raw data into a common format. Each contributed dataset is stored in an individual folder (Fig. 1) that contains a description of the protocol for data measurement (required, free format), a site information table (Table 1, required CSV file), a dataset information table (Table 1, required CSV file), the raw gas exchange data (free format), the reflectance data (free format), and the leaf sample details (free format).

The GSTI project uses a standardized approach to process and fit the raw reflectance and gas exchange data. This workflow is defined in a series of sequential data processing scripts written in the R programming language (R Core Team, 2024) and shown graphically in Fig. 1. Each R script has been designed to carry out a specific portion of the processing and model fitting workflow and shares a common design and function naming convention. This structure allows users to easily track the processing steps, from raw data to the final processed product. The repository offers several R tools and functions that can help visualize and check the quality of the data. A primary component of the processing chain includes the R functions to estimate photosynthetic traits from raw gas exchange measurements (*f.fit_ACi()* or *f.fit_One_Point()*, see section 2.2.2 "Photosynthetic gas exchange data and processing") and check the compliance of a new dataset with the repository requirements (*f.Check_data()*). A dataset creation guide is included in the repository (https://github.com/plantphys/gsti/wiki/Dataset-creation-guide).

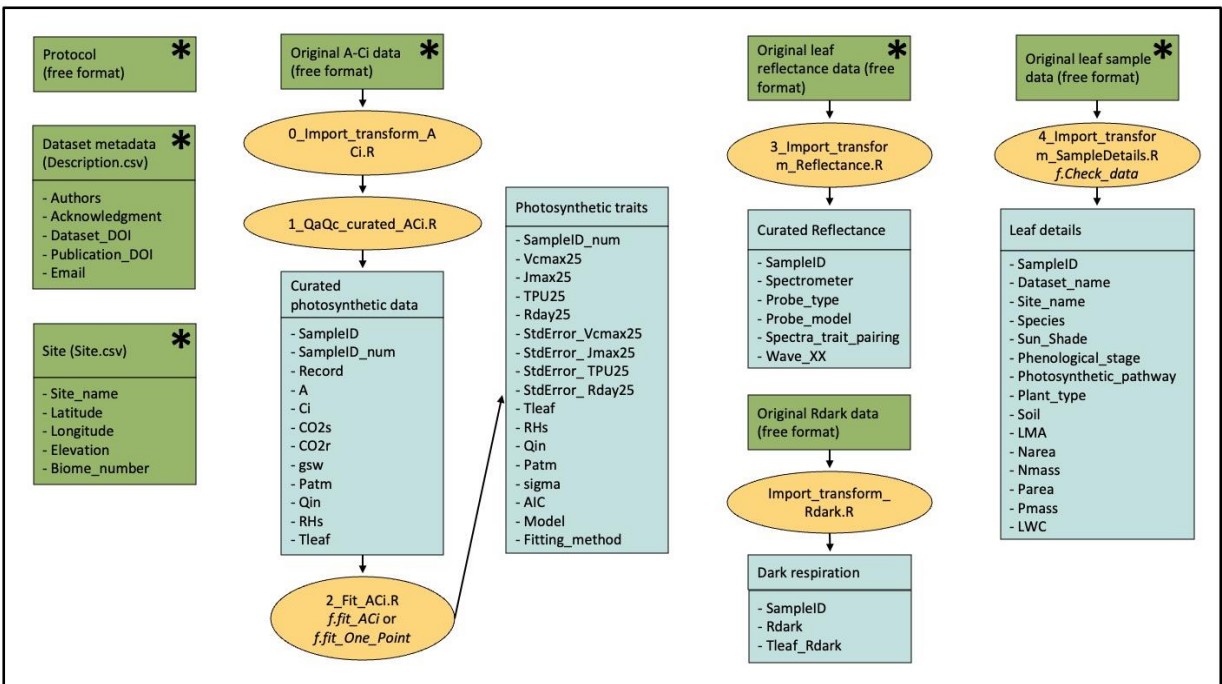

**Figure 1: Repository organization and process flow for each dataset. Each green rectangle represents one type of data. The asterisks represent the required original data. Each yellow circle represents one process associated with one R code that can be adapted for each dataset. Functions written in italics must be used to derive photosynthetic parameters (*f.fit_ACi()* or *f.fit_One_Point()*) and check the dataset compliance with the repository requirements (*f.Check_data()*). The lists within the blue boxes represent the lists of variables that need to be included in the database (e.g., Site variables) or produced by the R codes (e.g., Photosynthetic trait variables).**

**Table 1 List of variables included in the GSTI database**

| Variables | Definitions |
|---|---|
| **Dataset metadata** | |
| Authors | Dataset authors |
| Acknowledgment | Acknowledgement of funding and help to generate the dataset |
| Dataset_DOI | Dataset Digital Object Identifier |
| Publication_DOI | Digital Object Identifier of the main publication associated with the data |
| Email | Contact email for the dataset |
| **Site** | |
| Site_name | Site name |
| Latitude | Latitude, decimal; positive=north, negative=south |
| Longitude | Longitude, decimal; positive=east, negative=west |
| Elevation | Elevation, m |

| Biome_number | Biome number, based on Olson et al. (2001) classification. See online documentation: https://github.com/plantphys/gsti/wiki |
|---|---|

**Curated photosynthetic data**

| SampleID | Unique identifier of the sample leaf of a dataset as defined by the dataset authors in the raw gas exchange data |
|---|---|
| SampleID_num | Unique numerical identifier of the sample leaf of a dataset |
| Record | Gas exchange observation record number |
| A | Net $CO_2$ exchange per leaf area, µmol m$^{-2}$ s$^{-1}$ |
| Ci | Intercellular $CO_2$ concentration in air, µmol mol$^{-1}$ |
| CO2s | $CO_2$ concentration in wet air entering chamber, µmol mol$^{-1}$ |
| CO2r | $CO_2$ concentration in wet air inside chamber, µmol mol$^{-1}$ |
| gsw | Stomatal conductance to water vapor per leaf area, mol m$^{-2}$ s$^{-1}$ |
| Patm | Atmospheric pressure, kPa |
| Qin | In-chamber photosynthetic flux density incident on the leaf in quanta per area, µmol m$^{-2}$ s$^{-1}$ |
| RHs | Relative humidity of air inside the chamber, % (0-100) |
| Tleaf | Leaf surface temperature, °C |

**Photosynthetic traits**

| SampleID_num | Unique numerical identifier of the sample leaf of a dataset |
|---|---|
| Vcmax25 | Maximum rate of carboxylation at the reference temperature 25°C, µmol m$^{-2}$ s$^{-1}$ |
| Jmax25 | Maximum rate of electron transport per leaf area at the reference temperature 25°C, µmol m$^{-2}$ s$^{-1}$ |
| TPU25 | Triose phosphate utilization rate per leaf area at the reference temperature 25°C, µmol m$^{-2}$ s$^{-1}$ |
| Rday25 | $CO_2$ release from the leaf in the light at the reference temperature of 25°C, µmol m$^{-2}$ s$^{-1}$ |
| StdError_Vcmax25 | Standard error of Vcmax25 estimation, µmol m$^{-2}$ s$^{-1}$ |
| StdError_Jmax25 | Standard error of Jmax25 estimation, µmol m$^{-2}$ s$^{-1}$ |
| StdError_TPU25 | Standard error of TPU25 estimation, µmol m$^{-2}$ s$^{-1}$ |
| StdError_Rday25 | Standard error of Rday25 estimation, µmol m$^{-2}$ s$^{-1}$ |
| Tleaf | Average leaf surface temperature of the gas exchange measurements, °C |
| RHs | Average relative humidity of air inside the chamber, % (0-100) |
| Qin | Average in-chamber photosynthetic flux density incident on the leaf in quanta per area, µmol m$^{-2}$ s$^{-1}$ |
| Patm | Average atmospheric pressure, kPa |
| sigma | Standard error of the residuals of the fitted A-Ci curve, µmol m$^{-2}$ s$^{-1}$ |
| AIC | Akaike information criterion, µmol m$^{-2}$ s$^{-1}$ |
| Model | Photosynthetic limitations of the photosynthetic rate during the A-Ci curve: Ac, or Ac_Aj, or, Ac_Aj_Ap |
| Fitting_method | Method for estimating Vcmax25: One_point or ACi_curve |

**Dark respiration**

| SampleID | Unique identifier of the sample leaf of a dataset as defined by the dataset authors in the raw gas exchange data |
|---|---|
| Rdark | $CO_2$ release or $O_2$ consumption by the leaf in the dark at measurement temperature, µmol m$^{-2}$ s$^{-1}$ |

| | |
|---|---|
| Tleaf_Rdark | Leaf surface temperature, °C |

| **Curated reflectance** | |
|---|---|
| SampleID | Unique identifier of the sample leaf of a dataset as defined by the dataset authors in the raw gas exchange data |
| Spectrometer | Spectrometer model: SE PSR+ 3500, SVC HR-1024i, SVC XHR-1024i, ASD FieldSpec 3, ASD FieldSpec 4, ASD FieldSpec 4 Hi-Res, … |
| Probe_type | Type of probe used to measure the reflectance: Integrating sphere, Leaf clip, or, Imager |
| Probe_model | Probe model: SVC LC-RP, SVC LC-RP Pro, ASD Leaf Clip, … |
| Spectra_trait_pairing | Measurement pairing between gas exchange data and reflectance: Same, Similar, or Plant scale |
| Wave_XX | Reflectance at wavelength XX, % (0-100) |

| **Leaf details** | |
|---|---|
| SampleID | Unique identifier of the sample leaf of a dataset as defined by the dataset authors in the raw gas exchange data |
| Dataset_name | Dataset name |
| Site_name | Site name |
| Species | Species name |
| Sun_Shade | Leaf exposition in the canopy: Sun, Shade |
| Phenological_stage | Leaf phenological stage: Young, Mature, Old |
| Photosynthetic_pathway | Photosynthetic pathway: C3, C4, C2, CAM |
| Plant_type | Plant type: Wild, Agricultural, Ornamental |
| Soil | Soil type: Natural, Pot, Managed, Hydroponic |
| LMA | Leaf dry mass per unit area of fresh leaf, g m$^{-2}$ |
| Narea | Nitrogen content of leaf per unit area of fresh leaf, g m$^{-2}$ |
| Nmass | Nitrogen content of leaf by dry mass, mg g$^{-1}$ |
| Parea | Phosphorous content of leaf per unit area of fresh leaf, g m$^{-2}$ |
| Pmass | Phosphorous content of leaf by dry mass, mg g$^{-1}$ |
| LWC | Leaf water content, % (0-100) |

**2.2.2 Photosynthetic gas exchange data and processing**

The gas exchange data are processed using three steps (Fig. 1). In the first step, the raw gas exchange data are imported, compiled, and transformed into a standard format that follows the naming and metadata convention of (Ely et al., 2021). This step defines names and harmonizes the units of the gas exchange variables. In the second step, the data quality is analyzed and invalid data (abnormal $CO_2$ concentration, relative humidity, temperature or stomatal conductance data, and duplicated values) are removed. Due to variations in gas exchange equipment formats, data storage practices, and laboratory protocols, R scripts for both the first and second steps require customization for each new dataset to capture and properly handle idiosyncrasies associated with the data. After passing through the initial GSTI formatting steps, a curated dataset with a standard format for all datasets is generated (Table 1).

Finally, the gas exchange parameters are estimated from the curated gas exchange data files using the model fitting code (Fig. 1). This code uses the function "*f.fit_ACi()*" to estimate the parameters $V_{cmax25}$, $J_{max25}$, $TPU_{25}$, and their standard error at a reference temperature of 25°C by fitting the Farquhar, von Caemmerer, and Berry (FvCB) model of photosynthesis to the data (Bernacchi et al., 2001; Farquhar et al., 1980; Harley et al., 1992; Sharkey, 1985). The FvCB model assumes that the photosynthesis rate is the minimum of three potentially limiting rates: (1) the Rubisco-limited assimilation rate ($A_c$), which depends on $V_{cmax25}$, (2) the electron transport-limited assimilation rate ($A_j$), which depends on $J_{max25}$, and (3) the triose phosphate utilization (*TPU*) -limited assimilation rate ($A_p$). The transition from $A_c$ to $A_j$ and $A_j$ to $A_p$ is determined automatically to optimize the fitting. Depending on the range of $CO_2$ concentrations used to perform the A-Ci curves and the conditions of measurements, $A_j$ or $A_p$ do not necessarily limit $A$. The $A_j$ and $A_p$ limitations are only considered if they improve the fitting according to the AIC criterion. The $V_{cmax25}$, $J_{max25}$, and $TPU_{25}$ values are discarded when estimated with less than two points within the $A_c$, $A_j$, or $A_p$-limited regions. $V_{cmax25}$ can also be estimated with the one-point method (Burnett et al., 2019; De Kauwe et al., 2016) using the function '*f.fit_One_Point()*'. The FvCB model intrinsically assumes infinite mesophyll conductance; thus, estimated parameters $V_{cmax25}$, $J_{max25}$ and $TPU_{25}$ represent apparent values based on intercellular as opposed to chloroplastic $CO_2$ concentration.

### 2.2.3 Dark respiration gas exchange data and processing

In addition to fitted gas exchange parameters, the GSTI database also includes dark-adapted leaf respiration ($R_{dark}$). The leaf $R_{dark}$ could have been measured on the same leaf used for estimating photosynthetic capacity, or on independent leaves. The raw gas exchange data used for $R_{dark}$ estimation is imported and transformed following (Ely et al., 2021) (Table 1). Most studies measure $R_{dark}$ (i.e., negative $A$ values) as $CO_2$ release. We also chose to report $R_{dark}$ as a positive value. Some studies also measure the rate of $O_2$ consumption (Coast et al., 2019). Conversion from $O_2$ uptake to $CO_2$ release requires an estimate of the respiratory quotient (a number around 1) that should be detailed in the contributed dataset protocol. All data were normalized to a reference leaf temperature of 25°C using a common approach (Leuning, 2002).

### 2.2.4 Leaf reflectance data and processing

The leaf reflectance data were imported from the contributed format and units and processed into a standard format. The GSTI repository only includes measurements of leaf reflectance corrected against a white reference and measured with a black background. It does not support other measurements such as so-called "transflectance" data, measured with a white background. GSTI currently uses standardized reflectance measurements interpolated to 1 nm wavelength resolution, either by the data contributor before supplying the data or during the GSTI data pre-processing using a simple linear wavelength interpolation that can be added manually to the dataset-specific processing code, or using approaches provided in available R packages, e.g., spectrolab (Meireles and Schweiger, 2021). If specific corrections are deemed necessary based on the instrumentation, they should be applied before the data are provided. For some examples of pre-processing for use in spectra-

trait models, please refer to Burnett et al. (2021). Such procedures include calculating reflectance from measured irradiance and correcting for sensor biases across wavelength ranges.

### 2.2.5 Leaf sample and site information

Within GSTI, the latitude, longitude, and elevation of sites where the data were collected need to be specified (Table 1). The biome needs to be provided following Olson et al. (2001)'s list of 14 terrestrial biomes of the world, which we extended with

five managed environments (managed grasslands, field crop ecosystems, tree crop ecosystems, greenhouse ecosystems, and other managed ecosystems). The leaf information includes the light environment (sun or shade), leaf phenological stages (young, mature, old), species identity, plant type (wild, ornamental, agricultural), and soil type (natural, pot, managed, hydroponic). Associated leaf properties can also be added, including LMA, LWC, and the nutrient content of N and P on a per leaf area or mass basis. The addition of associated leaf properties to the GSTI database is encouraged but optional.

### 2.2.6 Dataset data quality checks

Most dataset quality verification is performed by the dataset authors in the preliminary steps of the data curation. In addition, the functions *f.fit_ACi()* and *f.fit_One_Point()* perform basic data quality checks to ensure that the photosynthetic data do not include abnormal $C_i$ or temperature values. If such values are found, the functions terminate and return an error. Finally, the function *f.Check_data()* is used to validate the format of the curated dataset and verify that all needed files are complete. It

also checks the range of values for most traits and warns users if they are outside the expected range, possibly due to unit issues. When this occurs, users are advised to check the data units and quality, but the function does not block them from adding the dataset to the database.

The standard error associated with the photosynthetic parameters (e.g., StdError_Vcmax25, StdError_Jmax25, Table 1) as well as the standard error of the residuals of the A-Ci curve-fitting (sigma, Table 1) can be used to filter the datasets and only

include the most reliable data for building spectra-trait models. For this purpose, the variable "Spectra_trait_pairing" is also important as it indicates whether or not the spectra and traits were derived on the same leaves or distinct but similar leaves. This information can be used for uncertainty analysis based on different sources of possible error.

### 2.3 Overview of the database and illustrative examples

The climate space covered by the GSTI database was evaluated using the mean annual temperature and precipitation of the

sites extracted using the Worldclim climate surface data (Fick and Hijmans, 2017) at the site positions and interpreted in the Whittaker biome classification (Ștefan and Levin, 2018; Whittaker, 1970).

For illustrative purposes and to show the extent of the GSTI database, we evaluated the correlations between $V_{cmax25}$ and other gas exchange variables ($J_{max25}$, $R_{dark25}$) as well as leaf biochemical, elemental, and morphological traits using Pearson correlation tests. We also evaluated the relationship between $V_{cmax25}$ and an estimate of the chlorophyll a and b content ($Chl_{index}$)

derived using spectroscopy. This estimate was derived using an index proposed by (Gitelson et al., 2003), calculated as the

ratio of reflectance in the spectral range of 750 to 800 nm ($R_{750\text{-}800}$) to the reflectance in the range 695 to 740 nm ($R_{695\text{-}740}$, Equation 1).

$$CHL_{index} = \frac{R_{750-800}}{R_{695-740}} - 1$$ (1)

One use of this GSTI database is to develop broadly applicable models of photosynthetic traits using reflectance data. Here, we used a partial least-squares regression (PLSR) modeling approach (Wold et al., 2001) to derive photosynthetic spectra-trait models following a previously described approach (Burnett et al., 2021). We focused on paired spectra-trait observations collected from the same leaves with full-range reflectance. Separate PLSR models were developed for $V_{cmax25}$, $J_{max25}$, $TPU_{25}$,

and $R_{dark25}$. The variables were square root-transformed and the database was divided into a calibration and a validation dataset by randomly selecting 80% of observations from each dataset for training and reserving the other 20% for external validation (stratified random sampling by source dataset). To train the PLSR models, 1000 random subsets of the training dataset were generated, each one containing 70% of the training observations. A PLSR model was fitted on each random subset and its performance was assessed on the remaining 30% of observations (internal validation). The number of components to use in

our final PLSR models was selected based on the predicted residual sum of squares (PRESS). We chose the smallest number of components that brought the PRESS one standard error away from the global minimum. The 1000 PLSR models were then applied to the external validation dataset and we calculated the mean prediction as well as the confidence interval of the prediction. The $R^2$ and root mean square error (RMSE) of prediction of the validation dataset were used to assess each model, as well as the %RMSE calculated as the ratio of RMSE to the range of variation of the trait of interest and expressed in percent

(0-100).

## 3. Results

The current release of the GSTI database contains 36 datasets (Fig. 2) and a total of 7525 observations of paired leaf reflectance and gas exchange data. Within the database, there are 4873 estimates of photosynthetic traits and 5075 estimates of $R_{dark}$. 2447 of these data have both photosynthetic traits and $R_{dark}$ measured on the same leaf. Of the data used to estimate photosynthetic

properties, 78% are A-Ci curves and 22% are one-point measurements. Among the A-Ci curves, three datasets were measured using the Dynamic Assimilation Technique (309 observations), and the others were measured using the common steady-state protocol. Most datasets were measured with full-range spectrometers (350 – 2500 nm, Fig. 3), and four were measured with shorter-range spectrometers (889 observations in total).

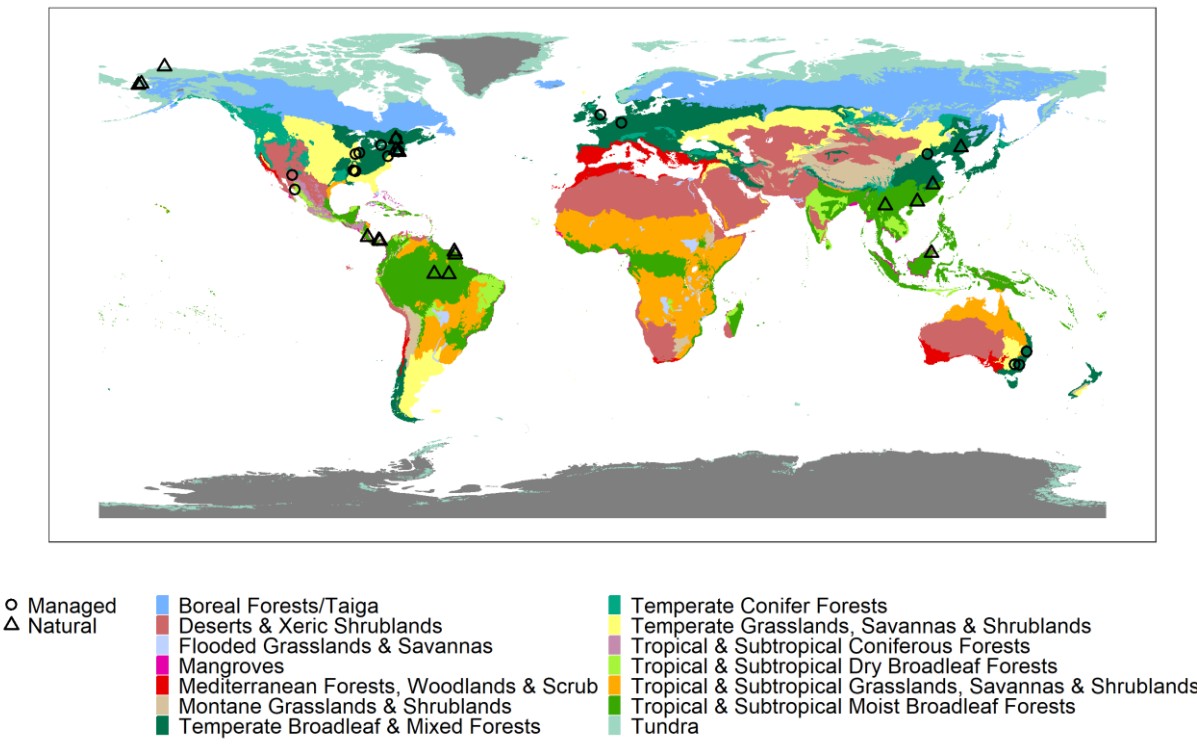

Boreal Forests/Taiga
Deserts & Xeric Shrublands
Flooded Grasslands & Savannas
Mangroves
Mediterranean Forests, Woodlands & Scrub
Montane Grasslands & Shrublands
Temperate Broadleaf & Mixed Forests

Temperate Conifer Forests
Temperate Grasslands, Savannas & Shrublands
Tropical & Subtropical Coniferous Forests
Tropical & Subtropical Dry Broadleaf Forests
Tropical & Subtropical Grasslands, Savannas & Shrublands
Tropical & Subtropical Moist Broadleaf Forests
Tundra

**Figure 2: Location of the datasets included in the database. Datasets represented with a circle were measured in a managed environment (greenhouse, growth chamber, field, etc.) whereas datasets represented with a triangle were measured in natural ecosystems. The map represents the 14 terrestrial biomes listed in Olson *et al.* (2001), updated by Dinerstein et al. (2017).**

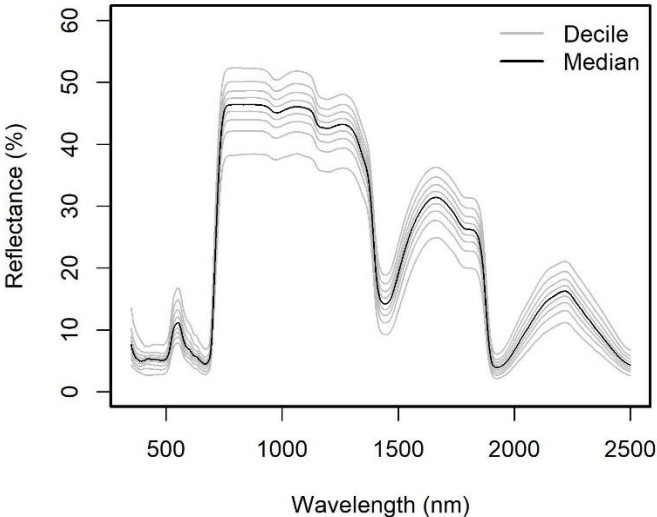

**Figure 3: Distribution of the reflectance of the database by decile. Only the full-range spectra were included (400 - 2500 nm).**

The observations were collected from 397 species, with the majority (293 species) coming from highly diverse ecosystems, including tropical and subtropical moist broadleaf forests (Figs. 2, 4, and 5). While the GSTI database also includes observations from tundra and temperate mixed broadleaf forests, it currently lacks data from several critical biomes, including Mediterranean forests, woodlands and scrub, other dryland ecosystems, montane grasslands and shrublands, and coniferous forests (Olson et al., 2001). Analysis of the climate space covered by the datasets within GSTI revealed that the sites were

mostly concentrated along a diagonal axis on the Whittaker plot (Fig. 4), missing the dry and warm biomes (bottom right corner of the plot) as well as the temperate wet environments (above the diagonal). A total of 23 species in the database are crops (Fig. 5), including wheat, rice, tomato, wine grapes, and tobacco. Although they belong to only a few species, the agricultural data represent 50% of the total observations in the database.

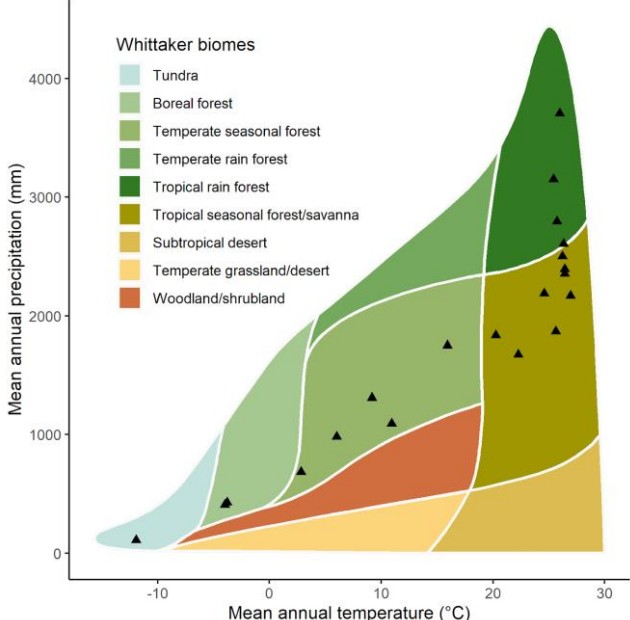

**Figure 4: Mean annual rainfall and mean annual temperature for the 20 natural sites included in the GSTI database plotted within the climatic boundaries of Whittaker's biomes.**

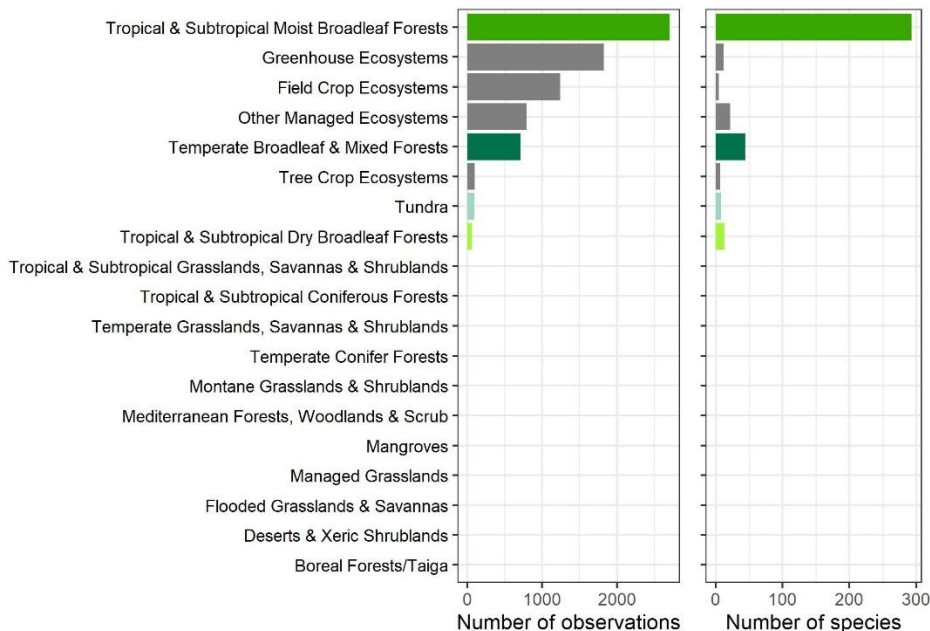

**Figure 5 Number of observations and species per biome. The list of biomes derives from Olson et al., (2001) list of 14 terrestrial biomes that we completed with 5 managed environment classes (managed grasslands, field crop ecosystems, tree crop ecosystems, greenhouse ecosystems, and other managed ecosystems).**

The distributions of the main traits included in the database are shown in Fig. 6. For $V_{cmax25}$, 95% of the values range from 3.7 to 186 µmol m$^{-2}$ s$^{-1}$, with an average of 67 µmol m$^{-2}$ s$^{-1}$ (Fig. 6a), while 95% of $R_{dark25}$ are in the range of 0.2 to 2.7 µmol m$^{-2}$ s$^{-1}$, with an average of 1.1 µmol m$^{-2}$ s$^{-1}$ (Fig. 6d). LMA was measured for 4553 observations and nitrogen content (Narea) for 3576 observations (Figs. 6e & 6f). LWC (Fig. 6h) and leaf P content (Parea, Fig. 6g) were measured less frequently, with 1849 and 785 observations, respectively. For LMA values, 95% fall within the range of 19 g m$^{-2}$ to 172 g m$^{-2}$ (Fig. 6e), while for Narea values, 95% fall within the range of 0.5 g m$^{-2}$ to 3.3 g m$^{-2}$ (Fig. 6f).

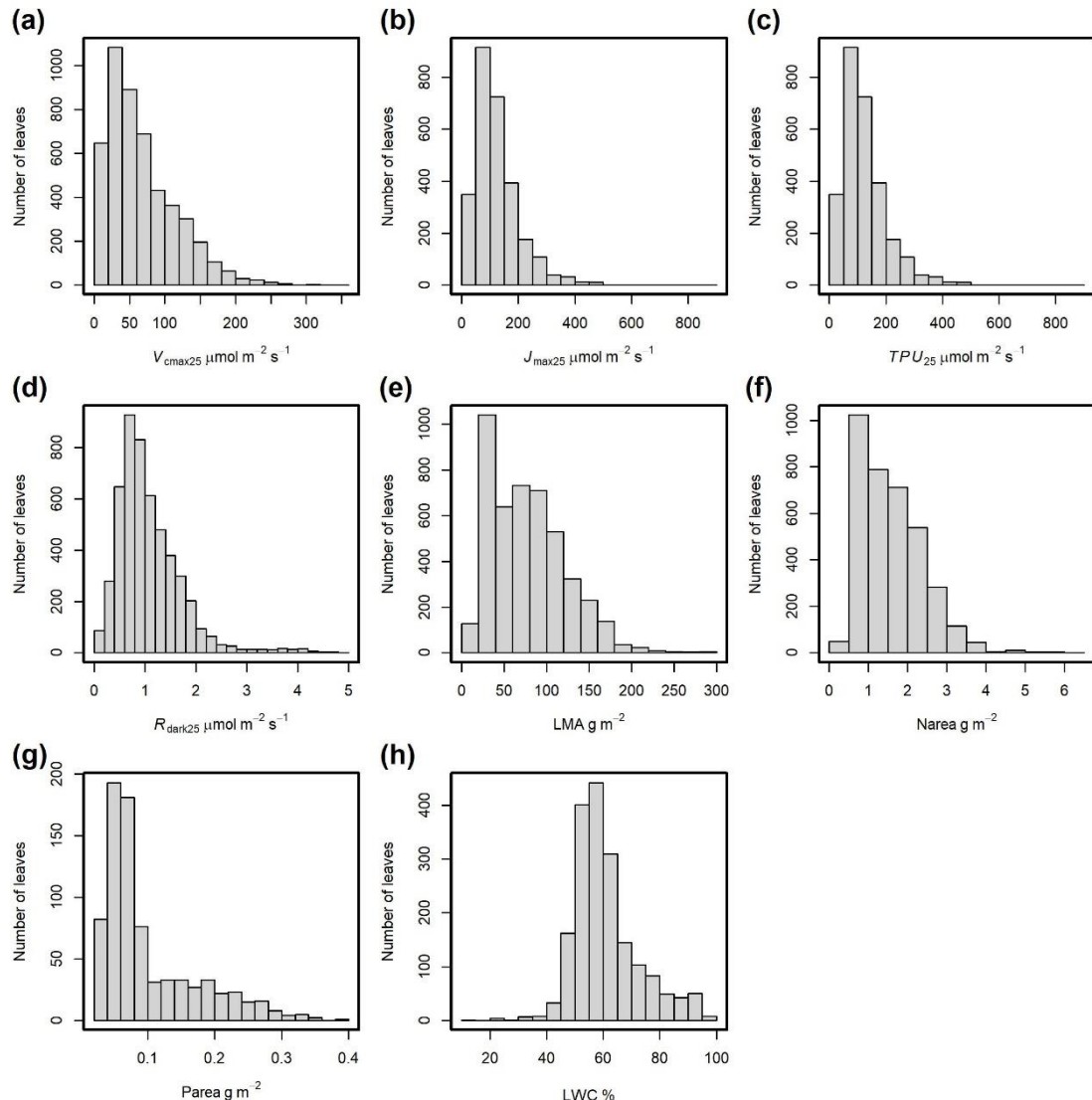

**Figure 6: Frequency distributions of observations for the main leaf traits.** a) Maximum carboxylation rate at 25 °C ($V_{cmax25}$). b) Maximum potential electron transport rate at 25 °C ($J_{max25}$). c) Triose phosphate utilization rate at 25 °C ($TPU_{25}$). d) Dark-adapted leaf respiration rate at 25 °C ($R_{dark25}$). e) Leaf dry mass per unit area of fresh leaf (LMA). f) Nitrogen content per surface area (Narea). g) Phosphorus content per surface area (Parea). h) Leaf water content (LWC).

Figure 7 illustrates the bivariate relationships between $V_{cmax25}$ and the other key variables including $J_{max25}$, $TPU_{25}$, $R_{dark25}$, LMA, Narea, and $Chl_{index}$, providing an overview of the scope and statistical properties of the current database. A strong correlation was observed between $TPU_{25}$ and $V_{cmax25}$ (r = 0.86, Fig. 7f) and between $J_{max25}$ and $V_{cmax25}$ (r = 0.94, Fig. 7d) with a $J_{max25}$:$V_{cmax25}$ ratio averaging 1.75. While $V_{cmax25}$ exhibited significant correlations with all other traits, the strength of these relationships was weak (|r| < 0.25), except for Narea, which demonstrated a moderate strength (|r| = 0.56, Fig. 7b).

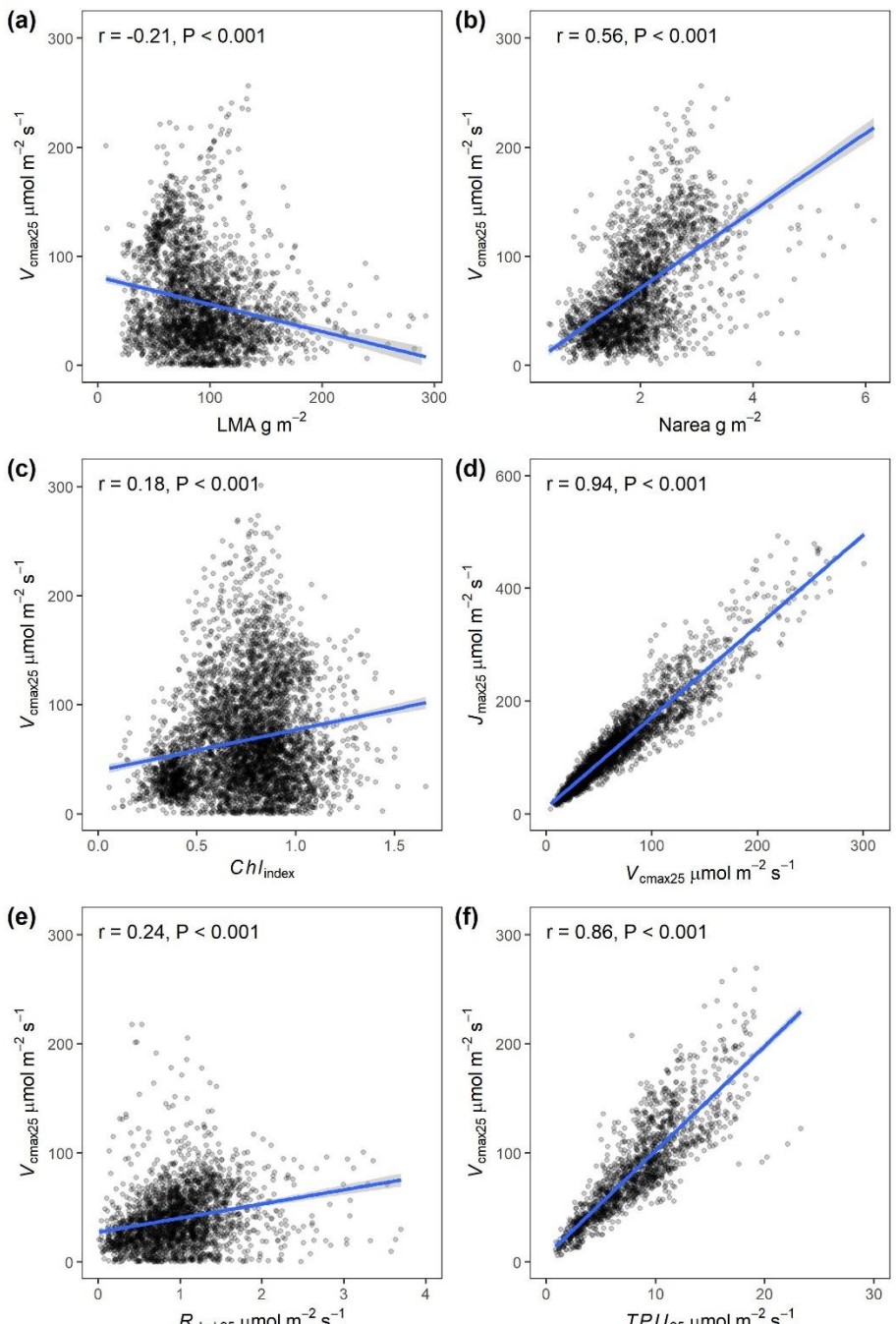

Figure 7: Scatter plot between the photosynthetic capacity ($V_{cmax25}$) and other leaf traits included in the GSTI database. a) Leaf mass per surface area (LMA). b) Nitrogen content on an area basis (Narea). c) Chlorophyll index derived from the reflectance data using Gitelson et al. (2003) index. d) Maximum potential electron transport rate at 25°C ($J_{max25}$). e) Dark-adapted leaf respiration rate at 25 °C ($R_{dark25}$). f) Triose phosphate utilization rate at 25 °C ($TPU_{25}$). The blue lines are linear regression fits.

A key goal of the GSTI project is to evaluate and test spectral models of leaf photosynthetic capacity and functional traits across a wide range of user-contributed datasets, spanning a wide range of ecosystems, species, growth conditions, and geographical locations. Therefore, in Fig. 8, we show an example of four spectra-trait models fitted between measured spectra and physiological traits including $V_{cmax25}$, $J_{max25}$, $TPU_{25}$, and $R_{dark25}$ using a PLSR approach. The performance of the spectra-trait models, evaluated on the 20% of observations from each dataset that were not used to train the models, were very strong

showing a R$^2$ of 0.77 for $V_{cmax25}$, 0.78 for $J_{max25}$, 0.79 for $TPU_{25}$ and 0.76 for $R_{dark25}$. The RMSE values of the models were always below 10% of the range of variation of the traits (%RMSE, Fig. 8).

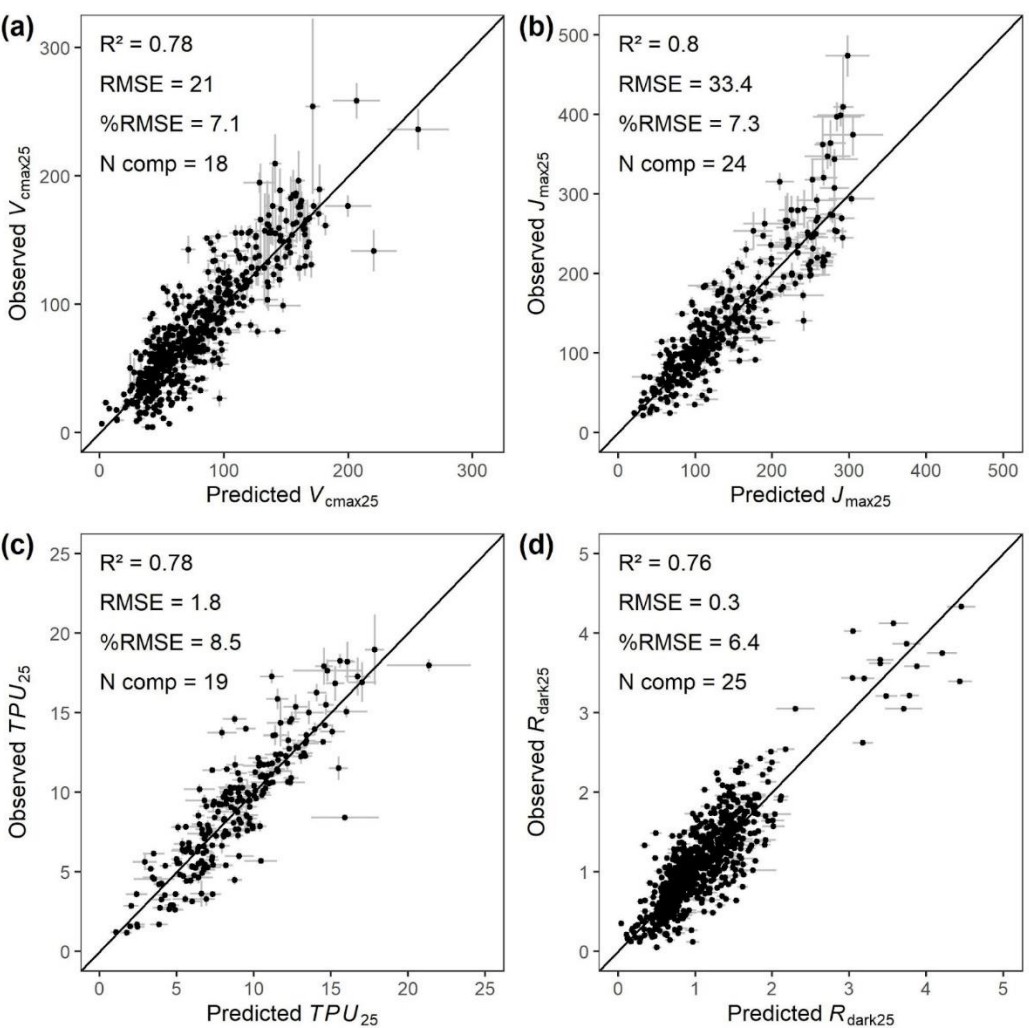

Figure 8: Observed photosynthetic properties obtained from gas exchange measurements vs reflectance-based partial least square

regression prediction. a) Maximum carboxylation rate of Rubisco ($V_{cmax25}$). b) Maximum potential rate of electron transport ($J_{max25}$).

**c) Triose phosphate utilization rate (*TPU*$_{25}$). c) Dark-adapted leaf dark respiration (*R*$_{dark25}$). The partial least square regressions were trained using 80% of the data from each dataset and validated with the remaining 20% of observations (points) following the best practice guide and protocol from Burnett *et al.* (2021). The validation points are shown with ± 95% confidence interval error bars. RMSE = Root Mean Square Error (μmol m$^{-2}$ s$^{-1}$), %RMSE is the root mean square error divided by the range of variation of the trait of interest in percent (0-100), and N comp is the number of components used in the partial least square regression.**

## 4. Discussion

The goal of the Global Spectra Trait Initiative is to enable a dramatic increase in the ability of the plant science community to estimate leaf traits using spectra-trait models developed with the richest possible datasets, in an open environment where all the data and tools to do so are freely available. The GSTI repository is focused on the preservation of paired leaf gas exchange and leaf reflectance data to facilitate the development and iterative improvement of spectra-trait models. It is designed to be collaborative, open access, and FAIR (findable, accessible, interoperable, and reproducible; Wilkinson et al., (2016)) following CC-BY 4.0 license protocols. A key feature of the GSTI repository is the preservation of the raw data and the provision of open workflows to process reflectance and gas exchange data, thereby ensuring uniform, reproducible data analysis and interpretation. The use of raw data also enables users to easily apply new techniques or alternative assumptions to their analysis and preserves the value of the repository for future unanticipated uses.

### 4.1 Data coverage

This initial GSTI version comprises 36 individual curated datasets, measured across 41 sites, spanning more than 390 species in arctic, temperate, subtropical, and tropical ecosystems. Given its breadth, this pooled dataset significantly expands upon previous photosynthetic and other spectra-trait modeling studies (Lamour et al., 2021; Serbin et al., 2012; Wu et al., 2025, 2019; Yan et al., 2021) and covers a wide range of climatological regions (Fig. 4). We hope that the expanding data coverage will ultimately enable the development of robust globally applicable models.

The spatial distribution of the current GSTI database is uneven. The majority of the datasets were measured on the American continent (Fig. 2), while Asia, Europe, and Australia have relatively limited data representation. African datasets are completely absent. Furthermore, of the 14 natural ecosystems categorized by Olson et al. (2001), only four are represented in the GSTI repository. These unrepresented biomes play key ecological roles and are likely to have distinct leaf spectral properties and associated traits, the absence of which likely limits the generalizability of spectra-trait models. The coverage of the four represented biomes is also incomplete and likely misses important functional groups and species. For example, few data cover needle-leaf coniferous plants (32 observations), none of which include full range reflectance spectra. Other plant groups such as ferns are also missing. Additionally, the database is biased towards a few dominant agricultural species, especially for *R*$_{dark}$ observations. These data limitations likely stem from several factors, including the high cost of spectrometers and gas exchange instruments. Additionally, challenges arise for certain plant types with very tiny or narrow leaves due to limitations of commercially available instruments designed for standard leaf sizes. Nevertheless, this underscores

the need to acquire observations spanning a broader spectrum of species diversity for increasing the leaf spectral and functional diversity as well as the development and testing of universally applicable photosynthetic spectra-trait models.

Tthe GSTI repository is currently limited to gas exchange data for leaves of the C3 photosynthetic pathway, the most prevalent photosynthetic pathway among plants. C4 plants, predominantly grasses, comprise less than 5% of known plant species (Sage, 2016), but they contribute to nearly a fifth of global photosynthesis (Luo et al., 2024). In addition, C4 crops like maize, millet, sorghum, and sugarcane account for nearly a quarter of the harvested area worldwide (Luo et al., 2024). Currently, spectra-trait studies for predicting C4 photosynthetic traits are primarily focused on maize (Heckmann et al., 2017; Wang et al., 2021; Yendrek et al., 2017). The crassulacean acid metabolism (CAM) pathway is another photosynthetic pathway used by around 6% of higher plants (Winter, 2019). To our knowledge, photosynthetic spectra-trait data have never been measured on such species, in part because measuring photosynthesis traits on these plants is also a challenge. The GSTI repository can therefore be expanded to accommodate other photosynthetic pathways in future updates. See Section 7 for how to contribute to future versions of the GSTI. We hope that highlighting the data gaps above will spur new data collection, improve the global coverage of the GSTI, and enhance its comprehensiveness.

## 4.2 Photosynthesis and dark respiration models

A single photosynthesis model (i.e., the same equations, kinetic constants and temperature response functions, and a standardized parameter estimation procedure) is used for all datasets within the GSTI database. This imposes a constraint on the datasets that can be added to GSTI; i.e., the raw gas exchange data must be supplied, not just the estimated photosynthetic parameters. Although this may limit the number of observations and datasets added to the GSTI repository, it avoids introducing biases between datasets that could arise from differences in parameterization of the photosynthesis model and the parameter estimation procedure. This standardization is beneficial for preserving the relationships between parameters, in particular between $V_{cmax25}$ and $J_{max25}$ (Fig. 7d), that depend upon the constants used in the photosynthesis model (Rogers et al., 2017a; Walker et al., 2014). Our approach to standardization within the GSTI database may also help reduce noise in trait-spectra relationships.

We acknowledge that other photosynthesis equations and parameterisations could be used. Fundamental processes of the C3 photosynthesis reactions are an active research area, and there are alternative formulations to the original model (Farquhar et al., 1980; Kumarathunge et al., 2019; Silva-Pérez et al., 2017; Yin et al., 2021). For example, other models consider a finite mesophyll conductance (Flexas et al., 2008), the cuticular pathway for gas transport between the leaf and the atmosphere (Lamour et al., 2022; Márquez et al., 2021), or a more mechanistic representation of electron transport rate (Johnson and Berry, 2021). Since the raw gas exchange data are saved in the GSTI database using a standard format (Ely et al., 2021), it is possible to reanalyze the data using other photosynthesis models. In fact, it is recommended to use the same set of equations and parameters for estimating photosynthetic traits and simulating photosynthesis, i.e., avoid mixing and matching equations (Rogers et al., 2017a). For instance, using an apparent $V_{cmax25}$ in a photosynthesis model with finite mesophyll conductance

would introduce errors. This consideration is also true when using traits estimated from spectral trait models. We recommend retraining the spectral trait model using the same photosynthesis model framework for both trait estimation and photosynthesis simulation. In addition, we have used a universal set of parameters to fit the gas exchange data. Better gas exchange fits may be obtained when species-specific (Sargent et al., 2024; Silva-Pérez et al., 2017) or environment-specific parameters
(Kumarathunge et al., 2019) are used, although it has yet to be demonstrated if this improves spectra-trait predictions.

Respiration, like photosynthesis, is a key physiological process that underpins plant growth and influences the global carbon budget and crop yields. However, unlike photosynthesis, which can be modeled using the FvCB photosynthesis equations (Farquhar et al., 1980), there is no comparable mechanistic model for respiration (Bruhn et al., 2022; Fan et al., 2024). This is partly due to our limited understanding of the complex metabolic processes underlying respiration and the difficulty in
estimating the complex networks of respiratory fluxes. In current crop growth and land surface models, leaf dark respiration is considered a temperature sensitive constant. These models use a range of temperature dependence functions (Huntingford et al., 2017; Niu et al., 2024). In the GSTI database, $R_{dark}$ and the leaf temperature during measurement are stored, facilitating the reuse of the data with alternative temperature response functions and parameterization.

### 4.3 Spectra-trait models

We have developed spectra-trait models for three photosynthetic traits ($V_{cmax25}$, $J_{max25}$, $TPU_{25}$) and $R_{dark25}$ as illustrative examples. These models were obtained using paired full-range spectral and trait observations on the same leaves. The predictive accuracy of these models aligns with previous studies (Barnes et al., 2017; Coast et al., 2019; Meacham-Hensold et al., 2019; Silva-Perez et al., 2018) despite incorporating a larger dataset spanning a wider range of species and environmental conditions and using datasets obtained with a range of instrumentation. This demonstrates the potential for such models to be
applied broadly, across diverse ecosystems and instruments. However, to optimize model performance for specific ecological contexts, the training dataset could be tailored to particular biomes or species. The training dataset used here exhibits an overrepresentation of certain species, especially agricultural ones, potentially biasing the model towards cultivated species and reducing its accuracy for other species.

A wide variety of statistical methods have been used to study the relationships between leaf spectra and traits. These include
PLSR, least absolute shrinkage and selection operator (LASSO), support vector machines (SVM), and deep learning (Burnett et al., 2021; Fu et al., 2019, 2022; Furbank et al., 2021; Ji et al., 2024; Vasseur et al., 2022). These approaches consider the specific features of reflectance spectra, i.e., a high dimensionality (hundreds of wavelengths) and a strong autocorrelation of the reflectance at each wavelength. They differ on how to reduce the signal dimension and on the form of the relationship between the signal and the trait of interest (linear or non-linear). The GSTI database offers a breadth of data to compare and
evaluate these different approaches.

An improved understanding of the mechanisms explaining the correlation between leaf reflectance spectra and photosynthetic performance is key to understanding the range of applications and the limitations of such models. Several hypotheses have been proposed to explain the relationship between leaf optical properties and photosynthetic traits such as $V_{cmax25}$. The

reflectance spectra are probably not strongly influenced by the quantity and activation state of rubisco that biologically determines $V_{cmax25}$, but more likely by a constellation of biochemical and structural leaf properties associated with photosynthetic performance that jointly shape the spectral signature associated with carboxylation capacity (Chadwick and Asner, 2016; Wu et al., 2019; Yan et al., 2021). The correlations between leaf reflectance spectra and nitrogen (Meacham-Hensold et al., 2019) or chlorophyll (Croft et al., 2017) content have been identified as important signals, although many other leaf structural and biochemical components are likely to play a role. Indeed, spectra-trait models often outperform trait-trait empirical relationships (Wu et al., 2025; Yan et al., 2021), even when multiple leaf traits are used for prediction. Since rubisco plays a role in carbon assimilation across all three photosynthetic pathways (C3, C4, and CAM), including these pathways in future analysis could be informative because the biochemical limitations on assimilation, leaf anatomy and elemental composition are different and will provide new axis of plant trait variation.

## 5. Conclusions

The Global Spectra-Trait Initiative (GSTI) is a collaborative and open-access database designed to facilitate the development and improvement of spectra-trait models for estimating leaf traits, focusing on photosynthetic capacity. The initial release of GSTI includes data from over 390 species and 41 sites, encompassing more than 7500 observations and covering a wide range of environmental conditions and plant functional types. It dramatically increases the data available for the plant science community. Furthermore, the standardized approach used in GSTI ensures uniform and reproducible data processing and interpretation, maximizing the reuse of data and facilitating ongoing refinement of spectra-trait models as new datasets are incorporated. Future developments of the GSTI will focus on expanding data coverage, incorporating data from under-represented biomes and plant functional types.

## 6. Data and code availability

The GSTI data and code are available in the public GitHub repository at https://github.com/plantphys/gsti, and published versions of GSTI are released to ESS-DIVE (https://data.ess-dive.lbl.gov/datasets/doi:10.15485/2530733, Lamour et al., 2025).

## 7. How to contribute to future versions of the GSTI

We encourage the community to contribute new datasets to expand the scope and utility of the GSTI project. To ensure consistency and maintain data quality, contributions should adhere to the standards and guidelines outlined in this paper. Detailed instructions for contributing datasets, including formatting specifications and submission procedures, are available in the project's GitHub repository: https://github.com/plantphys/gsti.

## Author contributions

Julien Lamour, Alistair Rogers, and Shawn Serbin conceptualized the project and supervised its realisation. Julien Lamour developed the database structure and the model code with inputs from Shawn Serbin. The manuscript was drafted by Julien Lamour, Shawn Serbin, and Alistair Rogers with input from Onoriode Coast, Shan Kothari, Fengqi Wu, and Daryl Yang. It was further revised by Kelvin T. Acebron, Nicolas Barbier, Carl Bernacchi, Angela C. Burnett, Joshua S. Caplan, Kim S. Ely, Jean-Baptiste Ferret, Claire Fortunel, Peng Fu, Bruno O. Gimenez, Troy Magney, Adam R. Martin, Milagros Rodriguez-Caton, Sheng Wang, Jin Wu, and Zhengbing Yan. All the authors were involved in the data acquisition, curation, and formatting of at least one dataset used in GSTI.

## Acknowledgements

This work was primarily supported by the Next Generation Ecosystem Experiments-Tropics project funded by the US Department of Energy, Office of Science, Office of Biological and Environmental Research, through U.S. Department of Energy Contract No. DE-AC02-05CH11231 to Lawrence Berkeley National Laboratory and by NASA's Surface Biology and Geology Mission, Biodiversity and Ecological Conservation program, and The Biospheric Sciences Laboratory (Code 618) at NASA Goddard Space Flight Center.

Additional support was provided by: a Discovery Grant from the Natural Sciences and Engineering Council of Canada and the University of Toronto Sustainable Food and Farming Futures Cluster; the Black Rock Forest Consortium through the David Redden Conservation Science Fund; an Oxford John Fell Fund; the ANR (the French National Research Agency) project BIOCOP (ANR-17-CE32-0001-01); NASA grants #80NSSC21K1707 and #80NSSC24K0135; the Bill & Melinda Gates Foundation, Foundation for Food and Agriculture Research and the UK Foreign, Commonwealth & Development Office under grant no. OPP11722157, and by Bill & Melinda Gates Agricultural Innovations grant investment ID 57248, the National Institute of Food and Agriculture, U.S. Department of Agriculture under award numbers 2018-67020-27934 and 2018-68005-27636; the Australian Research Council Centre of Excellence for Translational Photosynthesis (CE1401000015); the Next Generation Ecosystem Experiments-Arctic project funded by the US Department of Energy, Office of Science, Office of Biological and Environmental Research; the ANR (the French National Research Agency) under the "Investissements d'avenir" program with the references ANR-16-IDEX-0006, ANR-10-LABX-25-01, and the Amazonian Landscapes in Transition ANR project (ALT); the Spanish Government grant PID2022-140808NB-I00, and the Catalan Government grant AGAUR2023 CLIMA 00118; a multi-institutional research project (ANU2304-001RTX) on "Improving wheat yield through increases in heat tolerance of leaf carbon exchange" funded by the Australian Grains Research and Development Corporation (GRDC), Australian National University, University of Western Australia, University of Sydney, and University of New England Australia; GRDC-OzBarley EMCR travel award; University of New England tuition fee scholarships; and Future Food Systems CRC top-up scholarship (P2-035); a Discovery Grant from the Natural Sciences and Engineering Council of

Canada; by 80NSSC21K1713 U.S. Carbon Cycle Science Program NASA and Organization For Tropical Studies Lillian and

Murray Slatkin 521/572; a Discovery Grant from the Natural Sciences and Engineering Council of Canada; a USDA NIFA grant number 2022-67013-36205 (DRW); a Future Food Systems CRC grant (P2-023) on "Managing CO2 levels in controlled environment cropping systems to optimise yield" funded by the Future Food Systems Ltd, the University of New England Australia, and Tomato Exchange Pty Ltd; and Sally Muir Agricultural Research Award; the Novo Nordisk Foundation Starting Grant (NNF23OC0087612); the National Natural Science Foundation of China (#31922090), the HKU Seed Funding for

Strategic Interdisciplinary Research Scheme, and the Innovation and Technology Fund (funding support to State Key Laboratory of Agrobiotechnology); the National Natural Science Foundation of China (#32471573) and Key Talent Project of the State Key Laboratory of Vegetation and Environmental Change (LVEC-2023rc01). Michael Alonzo and Joshua S. Caplan were supported by funding from the National Science Foundation (award number 1951647). Urte Schlüter and Andreas P.M. Weber acknowledge funding by the Deutsche Forschungsgemeinschaft (German Research Foundation) under Germany's

Excellence Strategy EXC-2084/1 (project ID 390686111) and under the CRC TRR 341 (project ID 456082119). Peng Fu would like to acknowledge the support from Louisiana NASA EPSCoR Research Awards Program, which is sponsored by NASA & the Louisiana Board of Regents (BoR).

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
