# Peer review of "The Global Spectra-Trait Initiative: A database of paired leaf spectroscopy and functional traits associated with leaf photosynthetic capacity"

_Earth System Science Data, 2025_

## Referee Comment (RC1)

Reviewer Report: Manuscript essd-2025-213
Title: The Global Spectra-Trait Initiative: A database of paired leaf spectroscopy and functional traits associated with leaf photosynthetic capacity
Major Strengths

High Scientific Value: Establishes the first open-access global database (GSTI) systematically integrating leaf spectroscopy and photosynthetic functional traits, addressing a critical data gap for cross-species/environment spectral model development.

Methodological Standardization: Provides unified data processing workflows (R scripts) and parameter-fitting standards (e.g., FvCB model), ensuring data comparability and reproducibility.

Exceptional Data Scale: Covers 41 sites, 397 species, and >7,500 observations—significantly exceeding existing similar efforts.

FAIR Compliance: Open data (GitHub/ESS-Dive) under CC-BY 4.0 aligns with modern scientific data-sharing practices.
* * *
Required Revisions: Scientific and Logical Issues
1. Contradiction in Methodological Description

Issue (Page 10):
Original text: "We considered the mesophyll conductance infinite, therefore estimates of $V_{cmax25}$, $J_{max25}$ and are 'apparent' values..."
Problem: The phrasing "considered" inaccurately implies an active choice, while the FvCB model inherently assumes infinite $g_m$ .
Revision:

"The FvCB model intrinsically assumes infinite mesophyll conductance; thus, estimated parameters represent apparent values based on intercellular $CO_2$ concentration ($C_i$)."

2. Inadequate Discussion of Data Representation Bias

Issue (Pages 14–15, Fig. 5):

Data for temperate coniferous forests are minimal (32 observations, no full-range spectra), yet the text claims coverage of "temperate mixed broadleaf forests" without

highlighting this gap.

Africa is entirely unrepresented (e.g., savannas comprise ~11% of global vegetated area), potentially limiting model generalizability.
Revision: In Section 4.1 ("Data coverage"), quantify ecological significance of underrepresented biomes (e.g., African savannas, Mediterranean ecosystems) and assess impacts on model robustness.

**3. Ambiguous Model Validation Protocol**

Issue (Page 12):
PLSR validation uses "random selection of 80% for training and 20% for validation" but omits whether sampling was stratified by dataset. Global randomization risks data leakage if samples from the same dataset appear in both training/validation sets.
Revision: Clarify the sampling strategy (e.g., "stratified random sampling by source dataset") to prevent overestimation of model performance.

**4. Missing Figure Citations**

Issue (Page 16):
The description of trait correlations ("Figure 7 illustrates...") lacks a formal figure citation.
Revision: Insert "Figure 7" when first referenced:

"Figure 7 illustrates bivariate relationships between $V_{cmax25}$ and other traits..."
* * *
Language and Presentation Errors

Inconsistent Terminology (Page 6 vs. Table 1):

Text uses "leaf mass per area (LMA)", but Table 1 abbreviates it as "ALM".
Correction: Standardize to "LMA" throughout.

Ambiguous Units (Table 1):

"Wave_XX: Reflectance at wavelength XX, percent"
Correction: Specify as "Reflectance (fraction, 0–1)" or "Reflectance (%, 0–100)".

Incorrect Subscript (Fig. 6 Caption):

"TPUs    " → Correction: Use "TPU$_{25}$    " (consistent with text).

Typographical Error (Abstract):

"agricultrual" → Correction: "agricultural".
* * *
Additional Recommendations

Data Quality Control: Expand Section 2.2.6 to describe outlier handling (e.g., exclusion criteria) when f.Check_data() flags values outside expected ranges.

Roadmap for C4/CAM Data: In Section 4.1, specify plans/timelines to incorporate C4/CAM species (e.g., collaborations in progress).

Citation Updates: Replace preprint citations (e.g., Luo et al., 2024) with peer-reviewed versions where available, or label as "in review/preprint".
* * *
Decision
This work presents a valuable contribution to plant spectroscopy and functional ecology. However, revisions are required to address methodological clarity, data representativeness, and presentation consistency.
Recommendation: Minor Revision
* * *
Sincerely,

Invited Reviewer, ESSD

---

## Author Comment (AC1)

**The Global Spectra-Trait Initiative: A database of paired leaf spectroscopy and functional traits associated with leaf photosynthetic capacity**

**Response to editor and reviewers**

We thank the editor and reviewers for their time and constructive comments on our manuscript. We have revised the manuscript accordingly and have also addressed the editorial requests related to the author list and GSTI database references. We have also changed the colors of Fig 2 and Fig 5 to ensure better readability for colorblind readers, following editorial requirements.

We have added three authors to the author list (Robert Furbank, John Evans and Viridiana Silva-Perez) who were early contributors to the GSTI project, provided advice and data, but were inadvertently omitted from the original submission.

Our point-by-point responses to the reviewers are detailed below. The line numbers provided correspond to the version of the manuscript with track changes on.

Sincerely,

Julien Lamour, on behalf of all co-authors.

**Reviewer 1**

**R1.1** Major Strengths

High Scientific Value: Establishes the first open-access global database (GSTI) systematically integrating leaf spectroscopy and photosynthetic functional traits, addressing a critical data gap for cross-species/environment spectral model development.

Methodological Standardization: Provides unified data processing workflows (R scripts) and parameter-fitting standards (e.g., FvCB model), ensuring data comparability and reproducibility.

Exceptional Data Scale: Covers 41 sites, 397 species, and >7,500 observations—significantly exceeding existing similar efforts.

FAIR Compliance: Open data (GitHub/ESS-Dive) under CC-BY 4.0 aligns with modern scientific data-sharing practices.

Thank you for accepting to review our manuscript and for the constructive comments.

**R1.2** Contradiction in Methodological Description

Issue (Page 10): Original text: "We considered the mesophyll conductance infinite, therefore estimates of Vcmax25, Jmax25 and TPU25 are 'apparent' values..."

Problem: The phrasing "considered" inaccurately implies an active choice, while the FvCB model inherently assumes infinite gm.

Revision: "The FvCB model intrinsically assumes infinite mesophyll conductance; thus, estimated parameters represent apparent values based on intercellular $CO_2$ concentration (Ci)."

Agreed, we have changed the text as proposed (L 253).

**R1.3** Inadequate Discussion of Data Representation Bias

Issue (Pages 14–15, Fig. 5): Data for temperate coniferous forests are minimal (32 observations, no full-range spectra), yet the text claims coverage of "temperate mixed broadleaf forests" without highlighting this gap.

Africa is entirely unrepresented (e.g., savannas comprise ~11% of global vegetated area), potentially limiting model generalizability.

Revision: In Section 4.1 ("Data coverage"), quantify ecological significance of underrepresented biomes (e.g., African savannas, Mediterranean ecosystems) and assess impacts on model robustness.

Thank you for these suggestions. We have revised the second paragraph in Section 4.1 to address these points (see L 414 to 420 or texts below). The text highlighted in red below was added to the paragraph and strikethroughs were deleted.

The spatial distribution of the current GSTI database is uneven. The majority of the datasets were measured on the American continent (Fig. 2), while Asia, Europe, and Australia have relatively limited data representation. African datasets are completely absent. Furthermore, of the 14 natural ecosystems categorized by Olson et al. (2001), only four are represented in the GSTI repository. These unrepresented biomes play key ecological roles and are likely to have distinct leaf spectral properties and associated traits, the absence of which likely limits the generalizability of spectra-trait models.  The coverage of the four represented biomes is also incomplete and likely misses important functional groups and species. For example, few data cover needle-leaf coniferous plants (32 observations), none of which include full range reflectance spectra. Other plant groups such as ferns are also missing. Additionally, the database is biased towards dominant agricultural species, especially for $R_{dark}$ observations.

**R1.4** Ambiguous Model Validation Protocol

Issue (Page 12): PLSR validation uses "random selection of 80% for training and 20% for validation" but omits whether sampling was stratified by dataset. Global randomization risks data leakage if samples from the same dataset appear in both training/validation sets.

Revision: Clarify the sampling strategy (e.g., "stratified random sampling by source dataset") to prevent overestimation of model performance.

We have clarified this in the manuscript (L315)

**R1.5** Missing Figure Citations

Issue (Page 16): The description of trait correlations ("Figure 7 illustrates...") lacks a formal figure citation.

Revision: Insert "Figure 7" when first referenced: "Figure 7 illustrates bivariate relationships between $V_{cmax25}$ and other traits..."

We have confirmed that the initial citation of Figure 7 is correct. To improve clarity, we have also added specific citations to the figure's individual panels where they are referenced in the text (L 369 to 375).

**R1.6** Language and Presentation Errors

Inconsistent Terminology (Page 6 vs. Table 1): Text uses "leaf mass per area (LMA)", but Table 1 abbreviates it as "ALM".

Correction: Standardize to "LMA" throughout.

We have checked the text, tables and figures, including using the Ctrl + f command but didn't find any occurrence of "ALM" in the manuscript.

**R1.7** Ambiguous Units (Table 1):

"Wave_XX: Reflectance at wavelength XX, percent"

Correction: Specify as "Reflectance (fraction, 0–1)" or "Reflectance (%, 0–100)".

Thank you. Corrected.

**R1.8** Incorrect Subscript (Fig. 6 Caption):

"TPUs" → Correction: Use "TPU$_{25}$" (consistent with text).

We have checked the spelling of TPU in Fig.6 and elsewhere but haven't found any occurrence of "TPUs".

**R1.9** Typographical Error (Abstract):

"agricultrual" → Correction: "agricultural".

We have checked the spelling in the abstract but haven't found this typographical error.

**R1.10** Data Quality Control: Expand Section 2.2.6 to describe outlier handling (e.g., exclusion criteria) when f.Check_data() flags values outside expected ranges.

We have amended this paragraph to describe how outliers are handled (see L 284 to 290). The text highlighted in red below was added to the paragraph and strikethroughs were deleted.

In addition, the functions *f.fit_ACi()* and *f.fit_One_Point()* perform basic data quality checks to ensure that the photosynthetic data do not include abnormal $C_i$ or temperature values. If such values are found, the functions terminate and return an error.  Finally, the function *f.Check_data()* is used to validate the format of the curated dataset and verify that all needed files are complete. It also checks the range of values for most traits and warns users if they are outside the expected range, possibly due to unit issues. When this occurs, users are advised to check the data units and quality, but the function does not block them from adding the dataset to the database.

**R1.11** Roadmap for C4/CAM Data: In Section 4.1, specify plans/timelines to incorporate C4/CAM species (e.g., collaborations in progress).

We don't have specific timelines. However, we are open to collaborations and already mentioned this in the manuscript. See Section 7 "How to contribute to future versions of the GSTI" (L 515-519). We have added a sentence to Section 4.1 to highlight Section 7 How to contribute to future versions of the GSTI. (L 438-439).

**R1.12** Citation Updates: Replace preprint citations (e.g., Luo et al., 2024) with peer-reviewed versions where available, or label as "in review/preprint".

We have carefully checked the citations following your comments. The list of citations, including Luo et al. (2024), did not include preprints. The complete reference for Luo et al. 2024 which remain in the manuscript is pasted below.

Luo, X., Zhou, H., Satriawan, T. W., Tian, J., Zhao, R., Keenan, T. F., Griffith, D. M., Sitch, S., Smith, N. G., and Still, C. J.: Mapping the global distribution of C4 vegetation using observations and optimality theory, Nature Communications, 15, 1219, https://doi.org/10.1038/s41467-024-45606-3, 2024.

**Reviewer 2**

**R2.1** This GSTI dataset covers a diverse range of environmental conditions and species, allowing the development of robust modeling approaches. It makes a significant contribution to the scientific community. The discussion is also very informative, effectively highlighting existing gaps and potential areas for future improvements. Only a few major issues:

Thank you for your supportive review and for your comments.

**R2.2** Table 1. Are there any environment-related variables available, such as local weather data associated with the selected plants? These variables could provide valuable insights into environmental influences.

We agree that local weather data could be valuable but no, we did not collate environmental data associated with the paired leaf spectroscopy and functional traits. This is beyond the scope of this study.

**R2.3** Line 230: is it possible to create a figure showing the three data processing steps?

We have forgotten to include the reference to Figure 1 in this paragraph. We have now corrected this oversight (L 233).

**R2.4** Figure 7. Since the purpose is to show correlation, using correlation coefficient (e.g., R) instead of $R^2$ is better. In addition, RMSE should have units.

Agreed, we now present the Pearson correlation coefficient (r) which is indeed more appropriate. We have also removed the RMSE from this plot (See new Figure 7, L 375) and made corrections to the texts relating to Figure 7 (L 371-374).

Line 375: 'RMSE values were below 10%' might be a typo, RMSE has units and not in a percentage base.

We have amended the figure 8 caption to state the correct units for RMSE ($\mu$mol m$^{-2}$ s$^{-1}$) and clarified when we referred to RMSE (same unit as the variable) or %RMSE (in percent, 0-100) (L 323-324, 396-397). It is valuable to use %RMSE when comparing different variables.